# Sequence, Expression, and Anti-GCRV Function of the Ferritin from the Grass Carp, *Ctenopharyngodon idellus*

**DOI:** 10.3390/ijms23126835

**Published:** 2022-06-20

**Authors:** Tiaoyi Xiao, Dongfang Li, Hao Tang, Yijing Liao, Jun Zou, Yaoguo Li

**Affiliations:** Hunan Engineering Technology Research Center of Featured Aquatic Resources Utilization, Hunan Agricultural University, Changsha 410128, China; tyxiao1128@163.com (T.X.); dfli0927@163.com (D.L.); tanghao202206@163.com (H.T.); 13532107833@163.com (Y.L.); jzou@shou.edu.cn (J.Z.)

**Keywords:** ferritin middle subunit, cDNA and promoter, methylation level, immune response, anti-GCRV function

## Abstract

Ferritin possesses an immune function to defend against pathogen infection. To elucidate the immunity-protecting roles of ferritin from *Ctenopharyngodon idellus* (*Ciferritin*) against virus infection, the cDNA and promoter sequences of *Ciferritin* were determined, and the correlations between *Ciferrtin* expressions and promoter methylation levels were analyzed. In addition, the functional role of Ciferrtin on GCRV (grass carp reovirus) infection was assessed. The full-length cDNA of *Ciferritin* is 1053 bp, consists of a 531 bp open-reading frame, and encodes 176 amino acids. Ciferritin showed the highest sequence identity with the ferritin middle subunit of *Mylopharyngodon piceus* (93.56%), followed by the subunits of *Megalobrama amblycephala* and *Sinocyclocheilus rhinocerous*. Ciferritin contains a conserved ferritin domain (interval: 10–94 aa), and the caspase recruitment domain (CARD) and Rubrerythrin domain were also predicted. In the spleen and kidney, significantly higher *Ciferritin* expressions were observed at 6, 12, 24, or 168 h post GCRV infection than those in the PBS injection group (*p* < 0.05). The *Ciferrtin* expression level in the progeny of maternal-immunized grass carp was significantly higher than that in the progeny of common grass carp (*p* < 0.05). *Ciferritin* promoter methylation level in the progeny from common grass carp was 1.27 ± 0.15, and in the progeny of the maternal-immunized group was 1.00 ± 0.14. In addition, methylation levels of “CpG9” and “CpG10” loci were significantly lower in the progeny of maternal-immunized fish than those in the common group. Except for the “CpG5”, methylation levels of all other detected “CpG” loci negatively correlated with *Ciferritin* expression levels. Furthermore, the total methylation level of “CpG1–10” negatively correlated with the *Ciferritin* expressions. The *Ciferritin* expression level was significantly up-regulated, and the VP7 protein levels were significantly reduced, at 24 h post GCRV infection in the *Ciferritin* over-expression cells (*p* < 0.05). The results from the present study provide sequence, epigenetic modification and expression, and anti-GCRV functional information of Ciferritin, which provide a basis for achieving resistance to GCRV in grass carp breeding.

## 1. Introduction

Iron is an essential trace element for both the host and pathogen due to its role in energy metabolism, nucleic acid synthesis, and cell proliferation [1]. As an important intracellular iron storage protein, ferritin has been shown to store iron during times of iron overload and release iron in the condition of deficiency [2]. Ferritin consists of key domains, including a ferroxidase diiron center, an iron ion channel, and a ferrihydrite nucleation center [3,4]. As a hollow iron storage protein, it is composed of 24 subunits of heavy-chain and light-chain ferritins [5]. Specifically, three kinds of ferritin protein subunits were identified: heavy-chain homologous, which has a conserved ferroxidase site to facilitate the rapid oxidation and uptake of ferrous iron; light-chain homologous, which assists with the nucleation of the ferrihydrite iron core; and middle-type subunit, which possesses both a ferroxidase center and ferrihydrite nucleation center, [6]. Two kinds of channels, including the four-fold hydrophobic channel and the three-fold hydrophilic channel, were discovered in the protein shell of ferritin [7]. Iron enters ferritin to form 3.5–7.5 nm size iron cores mainly through the three-fold hydrophilic channels in the protein shell [8,9]. Ferritin keeps the redundant iron in a nontoxic and biologically available form and is a biomarker of iron-related disorders and infection [10].

Some viral pathogens promote their proliferation and diffusion by disturbing the iron homeostasis of the host cell [11,12]. Based on its iron storage property, ferritin has considerable effects on immune cell function, differentiation, and a host’s innate immune response to various pathogens [13,14]. In aquatic animals, ferritins have been reported to be involved in the immune response to pathogen infection. The transcriptional response of Atlantic salmon (*Salmo salar*) to infectious hematopoietic necrosis virus was investigated, and ferritin H was found to be involved in anti-viral replication via iron sequestering [15]. In hemocytes and hepatopancreas of red swamp crayfish (*Procambarus clarkii*), white spot syndrome virus (WSSV) and *Aeromonas hydrophila* infection induced the mRNA and protein expressions of ferritin, ultimately resulting in the inhibition of WSSV replication; moreover, the WSSV copy number was much larger in ferritin-silenced shrimp than in the control group [16]. The loss-of-function of ferritin in the red claw crayfish (*Cherax quadricarinatus*) led to a significantly higher expression of an envelope protein VP28 of WSSV, as demonstrated by the inhibited replication of WSSV after the transfection of recombinant ferritin protein [17]. Negative roles of ferritins in anti-disease were also observed in other fish species. For instance, in sea bass, recombinant ferritin-H function in the immune-suppression reaction of CXCR4, which is detrimental to innate immunity [18]. In addition to fishes, the ability of ferritin to enhance immune function is widely reported in other species [19,20].

The functional mechanisms of ferritins involved in pathogen infection immune reactions have been partially clarified. The host could use an iron withholding mechanism to restrict the availability of this essential nutrient to the invading pathogens. However, pathogens use various strategies to overcome this host defense. For example, WSSV protein kinase 1 interacts with the shrimp ferritin to prevent the ferrous ions binding of apoferritin [21]. Ferritin overexpression results in reduced expressions of MyD88-IRAK4, NF-κB, and TNFα promoter activity [22]. Ferritin also regulates the expressions of hepcidin through the NLRC5/MHC I/β2M axis and affects the adhesion of *Aeromonas hydrophila* to host cells by changing the expression of extracellular matrix proteins, integrin β1 and fibronectin [10].

The grass carp is one of the most widely cultured fish species in China, with its high production and remarkable economic contribution to agriculture. However, the hemorrhagic disease caused by grass carp reovirus (GCRV) threatens the sustainability of the grass carp farming industry. The disease resistance of grass carp individuals varies, and the identification of the immune molecule and its anti-GCRV function is pivotal for the molecular-assisted breeding of disease-resistant strains of grass carp. Female parents vaccinated with a GCRV-attenuated vaccine could produce a GCRV-resistant progeny, and we found that protein levels of Ciferritin were significantly higher in eggs of maternal-immunized grass carp than those of common fish [23]. In the present study, the full-length cDNA and promoter sequences of *Ciferritin* were identified, and their bioinformatic characteristics were analyzed. In addition, correlations between *Ciferrtin* mRNA and promoter methylation levels in grass carp individuals with different GCRV resistance were investigated. Furthermore, the immune functional effects of *Ciferrtin* on GCRV infection were evaluated. The results from this study help to deepen the understanding of fish ferritin involved in the viral infection process, providing a potential molecular resource for grass carp anti-disease breeding.

## 2. Results

### 2.1. Full-Length cDNA of Ciferritin

The full-length cDNA sequence of *Ciferritin* (GenBank accession no. MK934175.1) is 1053 bp in length. It consists of a 531 bp open-reading frame, a 202 bp 5′-terminal untranslated region, and a 320 bp 3′-terminal untranslated region. The predicted Ciferritin protein encodes 176 amino acids. The calculated molecular weight of *Ciferritin* is 86.81 kDa, and the isoelectric point is 4.98. In the 3′ non-coding region, polyadenylation signal site “AATAAA”, and the unstable motif “ATTTA” were identified (Figure 1).

### 2.2. Sequence Alignment and Functional Structure of Ciferritin

The BLAST analysis showed that *Ciferritin* shared the highest sequence identity with the *Mylopharyngodon piceus* ferritin middle subunit (93.56%, KY926441.1), followed by those of *Megalobrama amblycephala* (90.26%, KP288029.1) and *Sinocyclocheilus rhinocerous* (87.02%, XM_016540013.1). Multiple amino acid sequence alignment showed that the Ciferritin contains a conserved ferritin domain (interval: 14–155 amino acids). There were 54 identical amino acid residues among ferritin proteins and 44 conserved residues in the ferritin functional domain (Figure 2). Tertiary structure models of ferritins were constructed (Figure 3). The confidence score for the Ciferritin model is 1.04, and the estimated TM score for the tertiary structure of Ciferritin is 0.86 ± 0.07, supporting the reliability of this model. Five α-helices existed in the Ciferritin, whereas no β-sheets were identified. Functional domains of Ferritin, CARD, and Rubrerythrin were all predicted in Ciferritin (Figure 3).

### 2.3. Phylogenetic Tree Construction

A phylogenetic tree was built based on the ferritin protein sequences. The phylogenetic tree indicated that the ferritin homologs fall into two groups. One group contains the piscine, mammalian, aves, and amphibian branches, while the other group consists of the reptilia branches. The Ciferritin first clustered with that of *M. piceus*, followed by those in cyprinids, such as *Megalobrama amblycephala* and *Cyprinus carpio* (Figure 4).

### 2.4. Ciferritin Expression Change after GCRV Infection

The expression levels of *Ciferritin* in response to GCRV infection and PBS treatment were detected. For the spleen, GCRV infection induced significantly higher *Ciferritin* expression levels at 12 and 168 h than those in the PBS injection group (Figure 5A). In the kidney, *Ciferritin* expression levels were significantly higher in the GCRV infection groups at 6, 12, and 24 h than those in the PBS injection group (*p* < 0.05) (Figure 5B).

### 2.5. Ciferritin Promoter Sequence

The promoter sequence of the *Ciferritin* gene was cloned by genome walking PCR. The *Ciferritin* promoter sequence was 845 bp in length (GenBank: OK539049). Possible methylation modification “CpG” loci in the promoter and cDNA regions of *Ciferritin* were shown by bold letters, and methylation levels of ten “CpG” loci were all detected. Around the ten “CpG” loci, binding motifs for transcription factors such as SPI1, SP1, BRD3, NRF1, POU5F1, and ZBTB17 were predicted (Figure 6).

### 2.6. Correlation between Ciferritin Expression and Promoter Methylation Level

The expression levels of *Ciferrtin* in the progeny originated from the maternal-immunized grass carp and the common grass carp, which were detected using qPCR. The *Ciferrtin* expression level in the progeny of the maternal-immunized group was significantly higher than that in the progeny of the common grass carp (*p* < 0.05) (Figure 7A). The methylation levels of ten detected “CpG” loci in the spleen tissues between the progeny of maternal immunized and ordinary grass carp were all compared. The total methylation level of the ten “CpG” loci in the progeny of the common grass carp was 1.27 ± 0.15, and the progeny of the maternal-immunized grass carp was 1.01 ± 0.14. Methylation levels of both “CpG9” and “CpG10” were significantly lower in the progeny of the maternal-immunized fish than those of the common fish (Figure 7B). In addition, the methylation levels of “CpG2”, “CpG8”, “CpG9” and “CpG10” loci in the spleen tissues were significantly higher than those of other loci in the progeny of maternal-immunized and ordinary grass carp. Correlations between methylation levels of the ten detected “CpG” loci and mRNA levels of *Ciferritin* were analyzed. Except for the “CpG5”, methylation levels of all other “CpG” loci negatively correlated with *Ciferritin* expression levels, and the methylation levels of “CpG 1–2”, “CpG 3–5” and “CpG 6–9” negatively correlated with the expression levels of *Ciferritin*. The total methylation level of “CpG 1–10” was also negatively correlated with the *Ciferritin* expressions (−0.472, *p* = 0.238).

### 2.7. Anti-GCRV Effect of Ciferritin Overexpression

pEGFP-N1-Flag-Ferritin was successfully expressed in CIK cells (Figure 8A). At 48 h after *Ciferritin* overexpression, the CIK cells were infected by GCRV. At 24 h post infection, the expression levels of *Ciferritin*, *CiIFN1*, *CiMx*, *Vp2*, and *Vp7* were all evaluated by qPCR. The mRNA level of *Ciferritin* was significantly up-regulated in the overexpression experimental group at 24 h after GCRV infection. However, the *CiIFN1* and *CiMx* showed no significant differences between the *Ciferritin* overexpressing group and control group cells. For the viral content detection, the gene expression levels of *VP2* and *VP7* from GCRV were detected, and it was found that both *Vp2* and *Vp7* expressions were down-regulated in the *Ciferritin* overexpression CIK cells (Figure 8A–F). The protein level of GCRV VP7 level was also significantly lower in pEGFP-Ciferritin overexpressing cells (0.8182 ± 0.1066) than those in pEGFP-N1 transfected cells (1.2655 ± 0.3345) (*p* < 0.05) (Figure 8G,H).

## 3. Discussion

Ferritins are major iron storage proteins responsible for iron homeostasis and involved in pathogen infection immune response [24,25]. For lower order vertebrates, three kinds of subunits, including the heavy-chain, light-chain and middle-chain ferritins, were discovered [26]. Among the three types of ferritin subunits, the light-chain subunit is responsible for iron nucleation and mineralization, the heavy subunit facilitates the rapid oxidation and uptake of ferrous iron, and the middle subunit possesses both the functional properties of the heavy subunit and the light subunit [27]. Here, we identified both the *Ciferritin* middle-subunit cDNA and the promoter sequences for the first time. From the results of the phylogenetic tree and sequence–structure analysis, Ciferritin showed a closer relationship with homologous subunits of other fish species, and there are many conserved amino acids in the functional domains of ferritin. In addition, a caspase recruitment domain (CARD) and a Rubrerythrin domain have been predicted in the Ciferritin protein. The CARD associates with other CARD-containing proteins to play roles in apoptosis and inflammatory signaling [28]. Rubrerythrin-like molecules are the ancestor of ferritin and perform functions in oxidative stress protection via the catalytic reduction of intracellular hydrogen peroxide [29,30]. The sequence and functional domain characteristic results support that Ciferritin possesses a similar function attribute to other homologs.

As the pathogen of grass carp hemorrhage disease, GCRV can infect fish and cause significant economic loss in the aquaculture industry [31]. Finding gene resources with anti-GCRV functions is pivotal for high-resistance grass carp breeding. During infection, both the host and microbe need to access iron and avoid its toxicity, and in this regard, ferritin has emerged as a biomarker of pathogen infection, and variations in ferritin levels can be clinically relevant [1]. In this study, the expression changes in *Ciferrtin* in response to GCRV infection were investigated. We found that *Ciferritin* expressions in the immune tissues of the spleen and kidney were significantly enhanced at time points of 6 h, 12 h, 24 h, or 168 h post GCRV infection, suggesting that this molecule was involved in the immune response to grass carp hemorrhage disease. In other aquatic animals, ferritin is also reported to participate in anti-pathogen immune responses. The mRNA expression levels of scallop (*Chlamys farreri*) ferritin sharply increased in response to bacterial (*Vibrio anguillarum*) and viral (acute viral necrobiotic virus) challenges [13]. Ferritin could protect shrimp (*Litopenaeus vannamei*) and red claw crayfish (*Cherax quadricarinatus*) from WSSV infection by inhibiting virus replication or the deprivation of intracellular iron ions [17,32]. The purified recombinant ferritin was proven to help in reducing the mortality in shrimp (*Penaeus monodon*) infected with *Vibrio harveyi* [33].

To further clarify the functional effect of *Ciferritin* on GCRV infection, an overexpression experiment was successfully conducted, which found that the protein expressions of GCRV VP7 were reduced when combined with the expression enhancement of *Ciferritin*. VP7 is a composition protein of the GCRV outer capsid, which is critical for the assembly and disassembly of the reovirus [34,35]. Combined plaque and cytopathic effect-based TCID50 assays showed that the VP7 antibody was capable of neutralizing viral infectivity, suggesting that VP7 might be a dominating epitope [36]. This indicated that *Ciferritin* overexpression resulted in the inhibition of GCRV replication. In addition, we examined the expressions of *CiIFN1* and *CiMx* in grass carp, where no significant change in expression levels was monitored, suggesting that Ciferritin may not mainly implement its anti-GCRV function through the IFN pathway. A relationship between the *Ciferritin* mRNA expression and GCRV resistance of grass carp should be further elucidated.

Epigenetics plays a core role in affecting the expression of disease resistance-related genes across multiple generations and influencing the phenotype of individuals [37]. DNA methylation is the most well-understood epigenetic modification, and methylation levels of genes, such as tumor necrosis factor-like and arylhydrocarbon receptor nuclear translocator-2, contribute to pathogen resistance [38,39]. In addition, the methylation of the −534 CpG site of the RIG-I from grass carp had a close association with the resistance against GCRV and was significantly higher in susceptible individuals than in resistant individuals [40]. The epigenetic modification takes part in gene transcriptional regulation [41], and usually, higher methylation levels in the promoter region are negatively associated with gene expression levels [42,43]. From the correlation analysis results, we know that the methylation levels of almost every detected “CpG” locus and the total methylation level of the ten detected “CpG” loci are negatively correlated with *Ciferritin* mRNA expressions and in accordance with the negative relationship between promoter methylation and gene expression level. In addition, we compared the methylation levels in the promoter region of *Ciferritin* between progenies of the maternal-immunized grass carp and the common grass carp. We found that methylation levels of “CpG9” and “CpG10” in the progeny of maternal-immunized grass carp were significantly lower than those in the progeny of common fish, corresponding to significantly higher *Ciferritin* expressions in the progeny of maternal-immunized fish. Both innate and adaptive immune-relevant factors could be transferred from the mother to progeny in fishes and the expression levels of some immune genes up-regulated in the progeny of maternal-immunized fish [24,44]. The above results indicate that maternal immunization treatment might reduce the ferritin promoter methylation level in female parents, and the changed methylation mode may transfer to the progeny, ultimately corresponding to the up-regulated *Ciferritin* expression level and enhanced resistance.

Around the “CpG8” and “CpG9” loci, transcription factors binding motifs for SPI1, BRD3, POLR2A, NRF1, and SP1 were predicted, as well as POU5F1 and ZBTB17 around the “CpG10” locus. Transcription factor SP1 could bind to the regulatory element in the promoter region to promote gene transcription, whereas CpG island methylation weakens the binding resulting in the down-regulation of gene expression [45,46]. Upon virus infection, BRD3 promotes the recruitment of the IRF3/p300 complex to the promoter of Ifnb1, leading to the transcription and production of type I interferon [47]. NRF1 cooperated with DNA methylation to directly regulate the expression of multiple germ-cell-specific genes, and the conditional ablation of NRF1 dramatically down-regulated these genes [48]. For other transcription factors, the SPI1 could promote the transcription of the PARP9 gene [49], and the Pou5f1 could control the transcriptional activation of zygotically expressed genes in zebrafish embryos [50]. Transcription factors binding to the “CpG8–10” region may possess the ability to activate the expressions of *Ciferritin*. In combination, we can speculate that a lower methylation level in the “CpG8–10” loci in the progeny of maternal-immunized grass carp may benefit their binding to corresponding transcriptional factors, and ultimately enhance *Ciferritin* expression. The detailed effects of *Ciferritin* promoter methylation levels change with its expression, and the anti-GCRV effect should be further investigated.

In conclusion, we obtained the full-length cDNA and the promoter sequence of the *Ciferritin* middle unit. Ciferritin showed sequence similarity with homologous from other fish species, and the predicted functional domains, such as Ferritin, CARD, and Rubrerythrin, indicate its core roles in immune reactions. The methylation levels of the “CpG9” and “CpG10” loci in the progeny of maternal-immunized grass carp were significantly lower than those in the progeny of common fish, corresponding to significantly higher *Ciferritin* expression levels in the progeny of maternal-immunized fish. Promoter methylation and *Ciferritin* expression levels are negatively correlated. Ciferritin actively responded to GCRV infection, and its overexpression could inhibit the replication of GCRV. The value of Ciferritin for high-resistance fish breeding should be further evaluated in grass carp individuals.

## 4. Materials and Methods

### 4.1. Experimental Fish and Sample Collection

Grass carp, with an average body weight of 21.5 ± 2.4 g and an average body length of 12.5 ± 0.85 cm, were purchased from the Xiangyin Fisheries Research Institute, Yueyang, China. All fish were maintained in tanks for two weeks at a constant temperature of 28 °C and fed twice daily at 3% of their body weight. For the in vivo experiment, the GCRV106 strain with 1.78 × 10^7^ 50% tissue culture infective dose (TCID50)/mL was obtained from the Yangtze River Fisheries Research Institute, Chinese Academy of Fisheries Science (Wuhan, China). An experimental group and a control group were established for the GCRV infection. Thirty individuals in the experimental group were intraperitoneally injected with 200 µL of GCRV 106. Those in the control group (30 individuals) were intraperitoneally injected with 200 µL of PBS. The fish were euthanized by an overdose of tricaine methanesulfonate (200 mg/L) before sampling. In each group, samples (the kidney and the spleen) from five individuals were separately taken at 0, 6, 12, 24, 72, and 168 h after treatment. In addition, a GCRV JX-0901 strain (1 × 10^3.625^ 50% tissue culture infective dose (TCID50)/mL) was used for the cell infection experiment. Finally, spleen tissues of progeny from maternal immunized grass carp (female parent was immunized with a GCRV attenuated vaccine 40 days before fertilization), and tissues of progeny from ordinary grass carp were sampled for *Ciferritin* mRNA expression level and promoter methylation level detection.

### 4.2. RNA Extraction and cDNA Template Synthesis

For full-length *Ciferritin* cDNA cloning, total RNA was extracted from *C. idellus* spleen tissue using the Total RNA Kit (Omega Bio-Tek, Inc., Norcross, GA, USA), according to the manufacturer’s instructions. The concentration of RNAs was measured using a spectrophotometer (Eppendorf BioSpectrometer Basic, Hamburg, Germany), and the RNA integrity was analyzed by 1.3% agarose gel electrophoresis. RNA with an OD_260/280_ value ranging from 1.8 to 2.0 was chosen for cDNA synthesis. The template synthesis for full-length cDNA cloning was completed according to the SMARTer RACE cDNA amplification kit (Clontech, Palo Alto, CA, USA). For the quantitative real-time PCR (qPCR) template, the RNAs from tissues were treated with DNase I, and then the cDNA was synthesized using the ReverTra Ace-first-strand cDNA synthesis kit (Toyobo, Osaka, Japan), as per the manufacturer’s instructions.

### 4.3. Full-Length cDNA Cloning of Ciferritin

Based on the conserved sequences of ferritins from other fish species, primers Ft-F1 and Ft-R1, Ft-F2 and Ft-R2 for partial ferritin cDNA cloning were obtained using Oligo 7.0 software (Table 1). Based on the cloned sequence, two gene-specific primers, the Ferritin F and Ferritin R were designed for 5′ RACE and 3′ RACE PCR, respectively. The total volume of the RACE PCR system was 50 µL, including 15.5 µL PCR grade H_2_O, 25 µL 2× Seq Amp Buffer, 1 µL Seq Amp DNA Polymerase, 2.5 µL 3′ or 5′ cDNA template, 5 µL Universal Primer A Mix, and 1 µL 10 pmol/µL primer Ferritin F or Ferritin R. The PCR program was as follows: 95 °C for 5 min, 35 cycles at 95 °C for 30 s, 68 °C for 30 s and 72 °C for 1.5 min; and 72 °C for 7 min. The PCR products were purified, ligated into the pTOPO-TA vector (Solarbio, Beijing, China), transformed into *Escherichia coli* DH5α cells, and sent to the Sangon Biotech Company (Shanghai, China) for sequencing.

### 4.4. Promoter Cloning of Ciferritin

Genomic DNA was extracted from grass carp muscle by a DNA Kit (Omega Bio-tek, Norcross, GA, USA), following the manufacturer’s instructions. Four promoter libraries were constructed using the Universal GenomeWalker 2.0 Kit (Clontech, Mountain View, CA, USA). To obtain the promoter sequence of *Ciferritin*, two rounds of PCR were performed. The total volume of the first-round PCR was 50 μL, including 5 μL 10× BD Advantage 2 PCR Buffer, 1 μL BD Advantage 2 Polymerase Mix (50×), 1 μL dNTP (10 mM each), 1 μL 10 pmol/L AP1 primer, 1 μL 10 pmol/L ferritin P1, 1 μL template from the constructed EcoRV, PvuII, StuI or DraI genomic library, and an added 40 μL of ddH_2_O. The cycling program was as follows: 7 cycles of 94 °C for 25 s, 72 °C for 3 min; 32 cycles at 94 °C for 25 s, 67 °C for 3 min; and 67 °C for 7 min. In the second PCR cycle, one microliter of the reaction product from the first-round PCR was diluted 30 times and used as the template, and the primers AP2 and ferritin P2 were used for genome walking PCR. The PCR program was as follows: 5 cycles at 94 °C for 25 s and 72 °C for 3 min; and 20 cycles at 94 °C for 25 s, 67 °C for 3 min; and 67 °C for 7 min. The PCR products were detected by 1.5% gel electrophoresis, transformed into DH5α cells, and sent for sequencing.

### 4.5. Bioinformatics Analysis

The full-length cDNA of *Ciferritin* was joined by DNAMAN software, and the open reading frame and the deduced amino acid sequence of *Ciferritin* were predicted using the ExPASy-Translate tool (https://web.expasy.org/translate/ (accessed on 16 January 2022)). The isoelectric point and molecular weight of deduced amino acid sequences were predicted using the ExPASy-ProtParam tool (https://web.expasy.org/protparam/ (accessed on 12 February 2022)), and the domains of Ciferritin protein were analyzed by the Simple Modular Architecture Research Tool (http://smart.emblheidelberg.de/ (accessed on 18 February 2022)). Multiple sequence alignments were performed in DNAMAN 7.0 software, and a phylogenetic tree was constructed using the MEGA 7.0 software. Transcription factor binding characteristics in the ferritin promoter region were predicted by AnimalTFDB 3.0 [51].

### 4.6. Ciferritin Expressions in Grass Carp after Infection and of Different Resistance

The *Ciferritin* levels were detected in fish treated with PBS and GCRV, and in progenies of different GCRV resistances from maternal immunized grass carp (with enhanced GCRV resistance) and ordinary grass carp, quantitative real-time PCR (qPCR) was performed on the CFX96 Touch Real-Time PCR Detection System (Bio-Rad Laboratories, Hercules, CA, USA). The amplifications were in triplicate with a total volume of 10 µL containing 1 µL cDNA template, 5 µL of TB Green Premix Ex Taq II (2×), 0.4 µL primer ferritin YF, 0.4 µL primer ferritin YR and 3.2 µL ddH_2_O. The PCR program was as follows: 95 °C for 10 min, followed by 35 cycles of 95 °C for 10 s, 60 °C for 10 s, and 72 °C for 10 s. The *β-actin* and *EF1α* were chosen as the reference genes, and the relative expression levels of the target genes were calculated by the 2^−ΔΔCt^ method. The primers for the reference gene *β-actin* were β-actin YF and β-actin YR, and the primers for *18S* were 18sYF and 18sYR (Table 1).

### 4.7. Promoter Methylation Level Detection

The *Ciferritin* promoter methylation level was detected using the MALDI-TOF mass array technique, and the primers of Ferritin MF and Ferritin MR were designed for promoter region amplification by Agena EpiDesigner (http://www.epidesigner.com/ (accessed on 10 April 2021)). The amplification reaction containing 1 μL 10× PCR Buffer, 1 μL 200 uM dNTPs, 0.2 μL 5U/μL HotStar Taq (Qiagen, Valencia, CA, USA), 0.2 μL 10 pmol/μL ferritin MF, 0.2 μL 10 pmol/μL ferritin MR (Table 1), and 1 μL 10 ng/uL bisulfite treatment DNA template. The reaction programs were: 94 °C for 4 min; 45 cycles of 94 °C for 20 s, 56 °C for 20 s, and 72 °C for 1 min, followed by 72 °C for 3 min. The unincorporated dNTPs were removed by adding 1.7 μL RNase-free ddH_2_O and 0.3 U of Shrimp Alkaline Phosphatase to 5 μL PCR products. The reaction was performed at 37 °C for 2 min, and 85 °C for 5 min. The following transcriptase digestion reaction, dilution of the mixture, and mass spectra data analysis were similar to the report by Li et al. [52].

### 4.8. Anti-GCRV Function of Ciferritin Overexpression in CIK Cells

For the overexpression experiment, the open reading frame of *Ciferritin* was PCR-amplified with primer pairs Ferritin OF and Ferritin OR (Table 1). The total volume of PCR reactions was 50 µL, including 1 µL TransStart FastPfu Fly DNA polymerase (TransGen, Beijing, China), 10 µL 5× TransStart FastPfu Fly Buffer, 1 µL primer Ferritin-OF, 1 µL primer Ferritin-OR, 4 µL dNTPs, 2 µL cDNA template (spleen-originated template for qPCR) and 31 µL ddH_2_O. The PCR programs were one cycle at 95 °C for 2 min, 40 cycles at 95 °C for 20 s, 60 °C for 20 s and 72 °C for 30 s, and 72 °C for 5 min. The pEGFP-N1-Flag vector (Clontech, Palo Alto, CA, USA) and ORF PCR product were digested with *BamH*I and *EcoR*I, separately. The recombinant vector pEGFP-N1-Flag-Ferritin was constructed by cloning the purified ferritin ORF products into the pEGFP-N1-Flag vector, and the recombinant vector pEGFP-N1-Flag-Ferritin was sequence-verified. The pEGFP-N1-Flag- Ferritin was introduced into CIK cells. The experimental procedures were as follows: CIK cells were seeded into 25 cm^2^ cell culture flasks for 24 h with a minimum essential medium containing 10% FBS. A mixture of 8 µg plasmid, 16 µL P3000 and 15 µL lipofectamine 3000 (Invitrogen, Carlsbad, CA, USA) with 500 µL Opti-MEM medium were transferred into cells. Then, the transfection reagent was replaced with a fresh minimum essential medium at 6 h after transfection. At 48 h after overexpression, the cells were infected by GCRV JX-0901, and cells were sampled at 24 h post GCRV infection. The mRNA level of *Ciferritin*, *CiIFN1*, *CiMx*, *VP2*, and *VP7* in pEGFP-N1-Flag-Ferritin overexpressed CIK cells were all detected by qPCR, and the primers are shown in Table 1. The relative GCRV Vp7 protein levels in cells at 24 h after GCRV infection were detected by Western blot, with a procedure similar to Li et al. [53]. For Vp7 protein detection, the Vp7 polyclonal antibody (diluted 1:1000 in PBS) and the secondary antibody HRP-conjugated goat anti-mouse IgG (Abclonal, Wuhan, Hubei, China, 1:2000 in 1 × TBST) were applied in the Western blot experiment. For the *β*-actin protein detection, the *β*-actin polyclonal antibody and goat anti-rabbit IgG were adopted.

### 4.9. Fluorescence Microscopy

Fluorescence microscopy was applied to detect the expression status of the pEGFP-N1-Flag-Ferritin vector. CIK cells were cultivated into 6-well culture plates (35 mm diameter for each well) at a density of 2 × 10^5^ cell/mL for 24 h and incubated until the cells reached approximately 80% confluence. The recombinant vector pEGFP-N1-Flag-Ferritin was transferred into cells, and at 48h after overexpression, the cells were observed under a fluorescence microscope (Fluorescence Microcopy Olympus IX53, Tokyo, Japan).

### 4.10. Statistical Analysis

To compare the differences in mRNA and protein levels of tissues and cells, a one-way analysis of variance followed by Duncan’s multiple range tests in SPSS Statistics 22.0 software were used. Correlations between *Ciferritin* mRNA levels and promoter methylation levels were analyzed using Pearson correlation analysis. A *p*-value lower or equal to 0.05 was considered statistically significant, and those lower or equal to 0.01 were termed extremely significant.

## Figures and Tables

**Figure 1 ijms-23-06835-f001:**
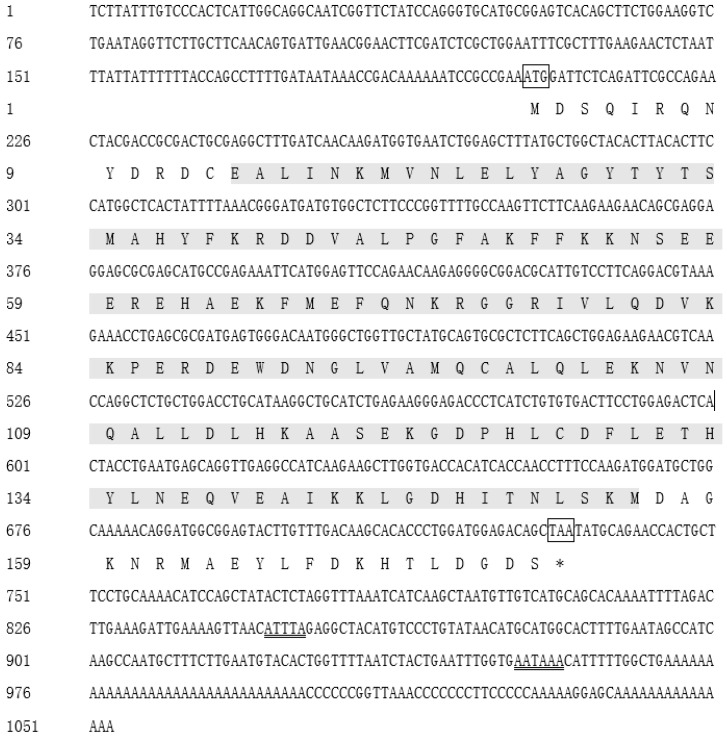
The nucleotide and deduced amino acid sequences of Ciferritin. The start codon is highlighted by the box, and the termination codon is marked with “*” and a box. The main domain of eukaryotic ferritin is marked with a gray shadow. The mRNA unstable motif (ATTTA) and polyadenylate termination signal (AATAAA) are double-underlined.

**Figure 2 ijms-23-06835-f002:**
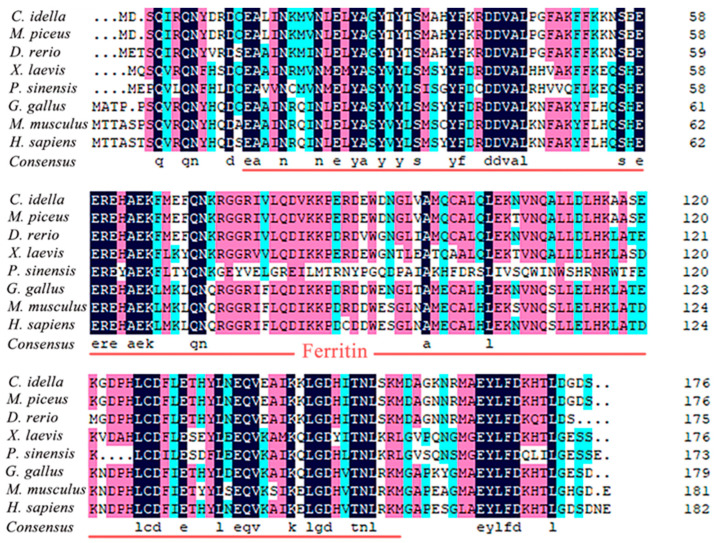
Multiple sequence alignment of Ciferritin with other homologs. The functional domain of ferritin was underlined.

**Figure 3 ijms-23-06835-f003:**
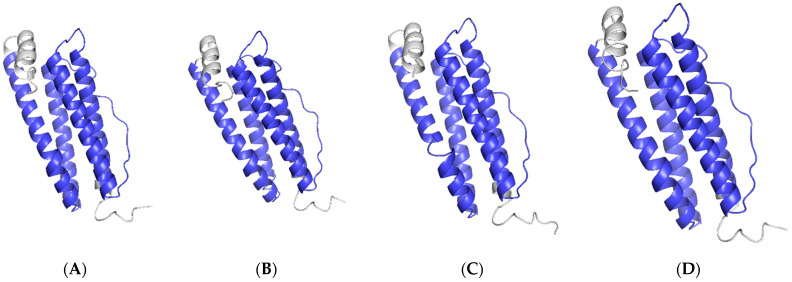
Tertiary structure of ferritin proteins. Ferritins from (**A**): grass carp (*Ctenopharyngodon idella*), (**B**): black carp (*Mylopharyngodon piceus*), (**C**): zebrafish (*Danio rerio*), (**D**): African clawed frog (*Xenopus laevis*); (**E**): Chinese soft-shelled turtle (*Pelodiscus sinensis*), (**F**): red junglefowl (*Gallus gallus*), (**G**): mouse (*Mus musculus*) and (**H**): human (*Homo sapiens*).

**Figure 4 ijms-23-06835-f004:**
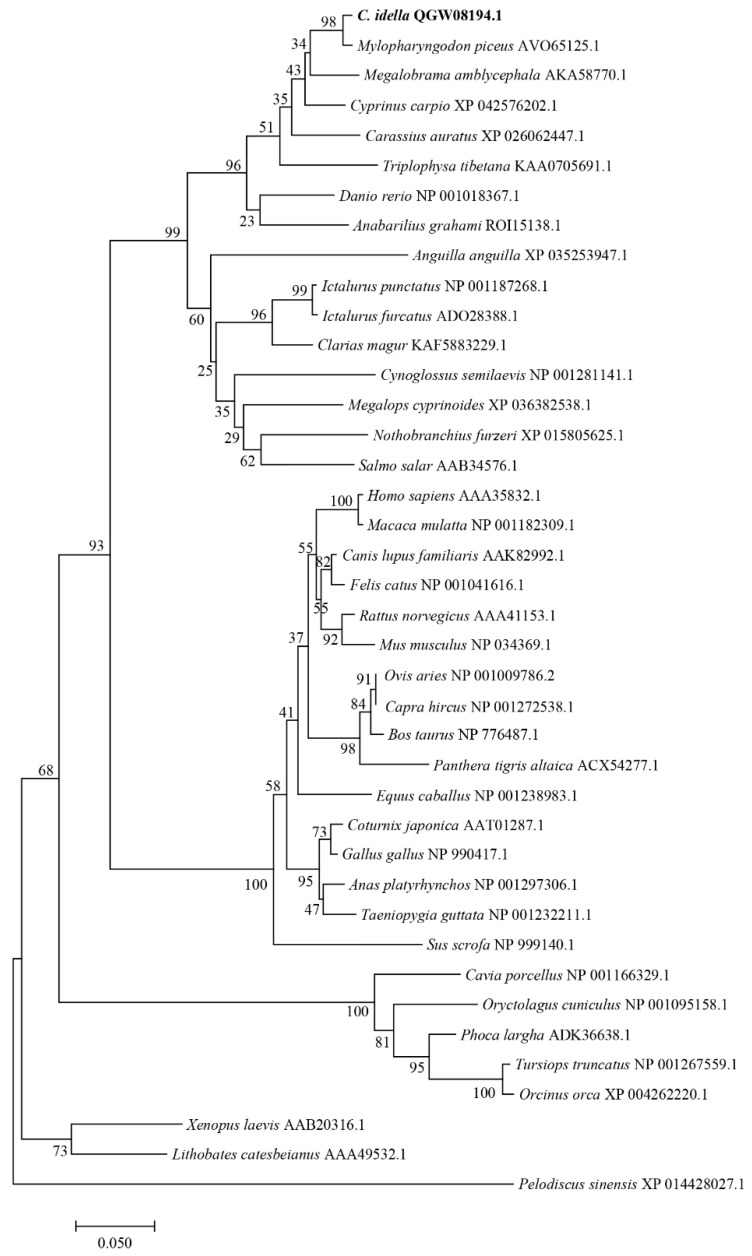
Phylogenetic tree of ferritins.

**Figure 5 ijms-23-06835-f005:**
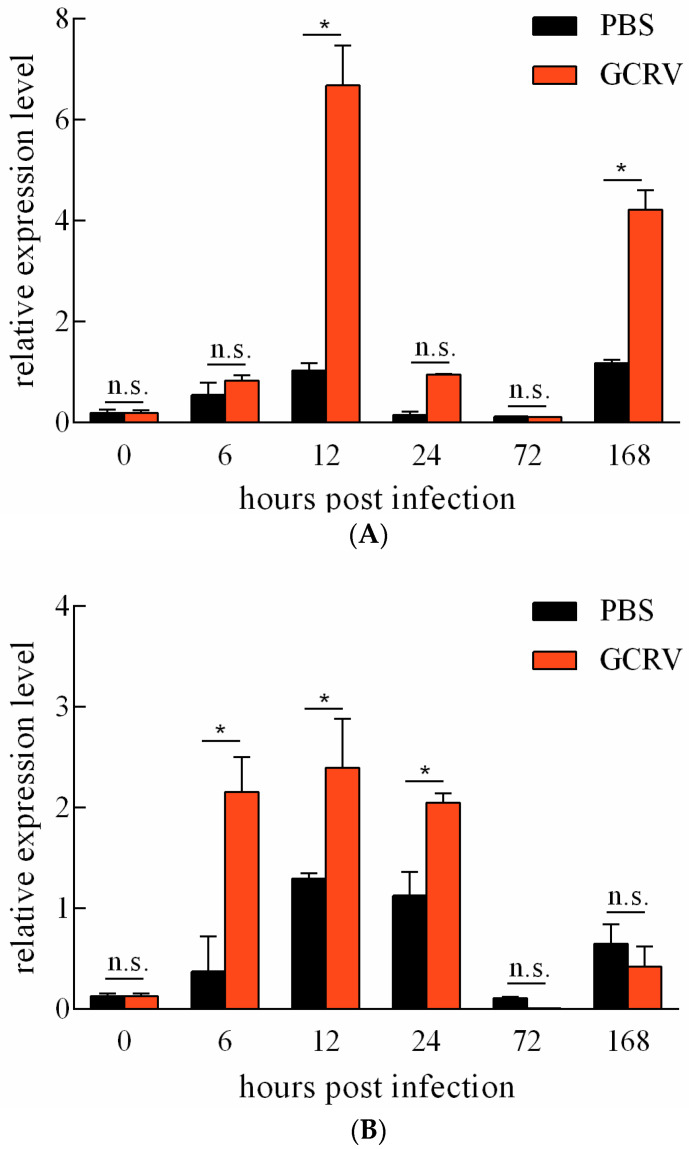
Expression level change in *Ciferritin* after PBS treatment and GCRV infection in the spleen (**A**) and the kidney (**B**). An asterisk (*) denotes a significant differences (*p* < 0.05) between PBS and GCRV treatment groups and n.s. indicates no significant difference (*p* > 0.05).

**Figure 6 ijms-23-06835-f006:**
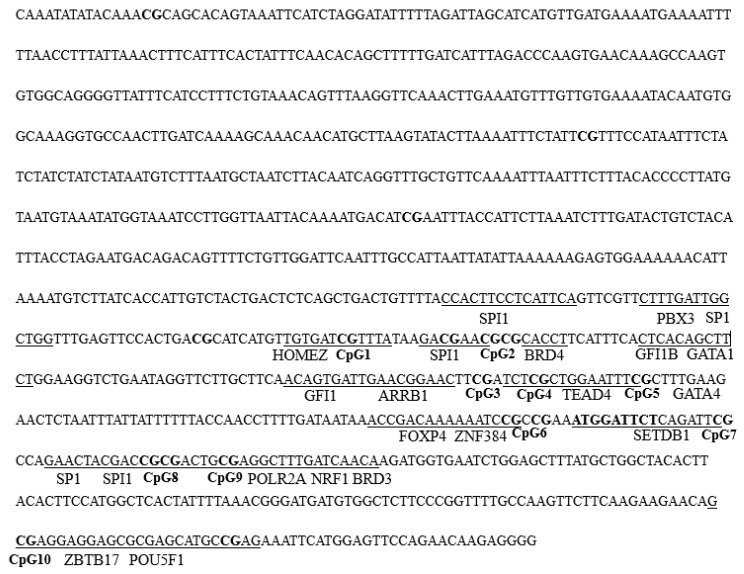
The *Ciferritin* sequence for DNA methylation detection. The bold letters indicate the “CpG1–10” loci, and the predicted transcription factor binding sites are underlined.

**Figure 7 ijms-23-06835-f007:**
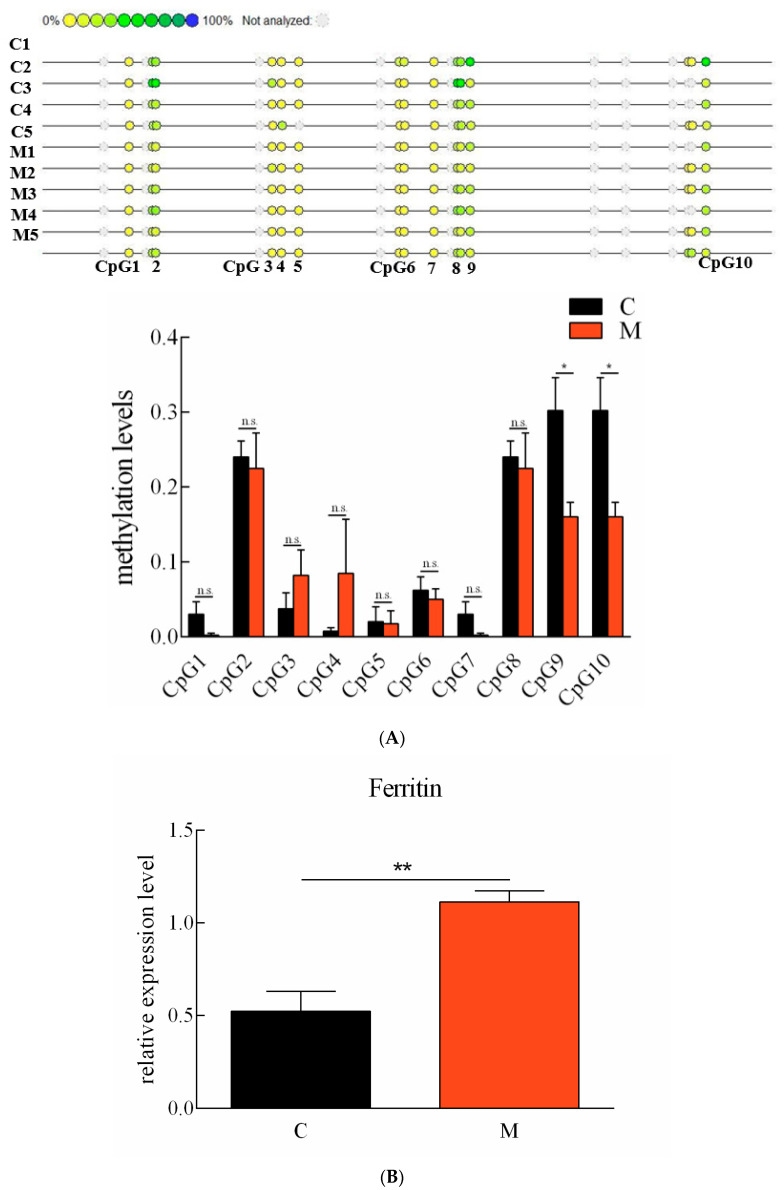
Methylation levels of “CpG” loci and mRNA expression levels of *Ciferritin* in grass carp individuals of different resistance. (**A**) Ferritin promoter “CpG” loci methylation levels, and (**B**) mRNA expression level comparison in the progeny of ordinary and maternal immunized grass carp. An asterisk (*) denotes a significant differences (*p* < 0.05), ** indicates *p* < 0.01, and n.s indicates no significant difference (*p* > 0.05). Letter C indicates progeny from ordinary grass carp, and M represents progeny from maternal immunized grass carp.

**Figure 8 ijms-23-06835-f008:**
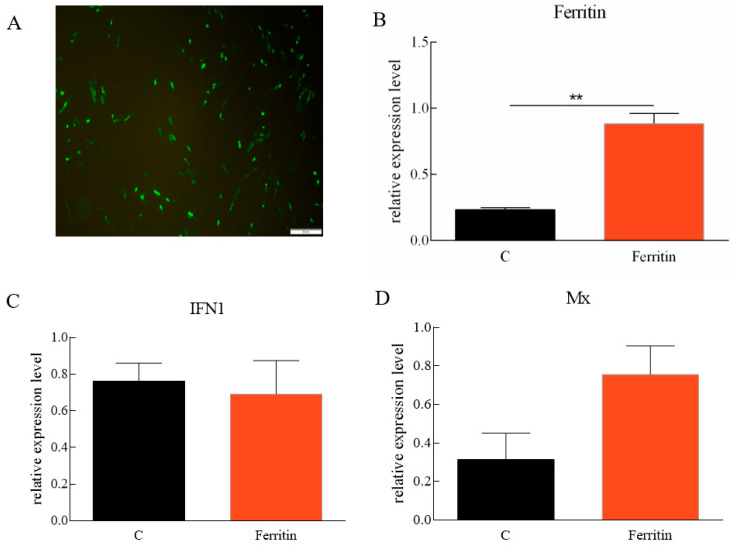
Effects of *Ciferritin* overexpression on immune reaction and GCRV replication. (**A**) Cell fluorescence of *Ciferritin* overexpression; (**B**) *Ciferritin* mRNA level; (**C**) *CiIFN1* mRNA level; (**D**) *CiMx* mRNA level; (**E**) *VP2* expression level; (**F**) *VP7* expression level; and (**G**,**H**) VP7 protein level. An asterisk (*) denotes a significant differences (*p* < 0.05), and ** indicates *p* < 0.01.

**Table 1 ijms-23-06835-t001:** The primers used in cDNA and promoter cloning, qPCR, and overexpression.

Primer Name	Primer Sequence 5′-3′	Usage
Ft-F1	AGATTCGCCAGAACTACGAAC	RACE
Ft-R1	GCTGCATGACAACATTAGCTT	RACE
Ft-F2	GAAGAACGTCAACCAGGCTCT	RACE
Ft-R2	ACATTCAAGAACGCATTGGCT	RACE
Ferritin F	CGTATTCCGACGTAGACTCTT	qPCR
Ferritin R	AAAGTTAACATTTAGAGGCTACA	qPCR
Ferritin P1	ATGAATTTCTCGGCATGCTCGCGCTCCT	Promoter cloning
Ferritin P2	AGTGTAGCCAGCATAAAGCTCCAGATTCACCA	Promoter cloning
Ferritin YF	TCTTCCCGGTTTTGCCAAGT	qPCR
Ferritin YR	TCATCGCGCTCAGGTTTCTT	qPCR
β-actin YF	GCTATGTGGCTCTTGACTTCG	qPCR
β-actin YR	GGGCACCTGAACCTCTCATT	qPCR
18sYF	ATTTCCGACACGGAGAGG	qPCR
18sYR	CATGGGTTTAGGATACGCTC	qPCR
IFN1 YF	AATGCTCTGCTTGCGAATG	qPCR
IFN1 YR	CCTGGAAATGACACCTTGG	qPCR
Mx YF	CGACCACAGAAGCATTGCAGA	qPCR
Mx YR	CCCTTCAGTGCCTTTATCCACCA	qPCR
VP2F	GAGCTTACCGGCGTCCTGAT	qPCR
VP2R	GGTCGGAGGCCATCGTGTAA	qPCR
VP7F	CCATGACACTCACGCACACG	qPCR
VP7R	GGCAAGCGAAGGTCAGGTTG	qPCR
Ferritin MF	aggaagagagAGAATGATAGATAGTTTTTTGTTGGA	Methylation detection
Ferritin MR	cagtaatacgactcactatagggagaaggctCATAAAACTCCAAATTCACCATCTT	Methylation detection
Ferritin OF	tcgagctcaagcttcgaattcATGGATTCTCAGATTCGCCAGA	Overexpression
Ferritin OR	cgtcatggtggcggcggatccGCTGTCTCCATCCAGGGTGTG	Overexpression

## Data Availability

The data presented in this study are available on request from the corresponding author.

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
