# Peer review of "Sequence, Expression, and Anti-GCRV Function of the Ferritin from the Grass Carp, *Ctenopharyngodon idellus"

_ijms, 2022, doi:10.3390/ijms23126835_

Round 1
Reviewer 1 Report
This is a straightforward and well designed study on the ability of the iron carrier ciferritin to modulate immune system activity and potentially provide resistance against a specific viral infection important in carp culture. There are many strengths of the study and manuscript. First the background for each aspect is presented well including fish ferritins, mechanisms of immune system enhancement in fish by iron, and nicely the economic need for economical effective immune enhancement to the viral infection. The scientific methodologies are clearly presented with relevant details and results presented well. The in vitro data is also presented well. One item that I would have strengthened the study is an in vivo model of viral infection and if this is available, should be presented. Additionally, the authors may consider adding only a few sentences that the ability of ferritin and iron to enhance immune function is not unique to fishes, it is a phylogenetically conserved physiologic effect.
Author Response
Thank you very much for the valuable comments. We add a few sentences to support the conserved physiologic effects of ferritin among animals of different classes, please see Line 69-70 in the “Introduction” part. We will perform in vivo model of viral infection experiment once the experimental conditions are completed in our future study.
Reviewer 2 Report
Dear Authors,
Re: Journal: IJMS (ISSN 1422-0067)
Manuscript ID: ijms 1769140
Type: Article
Title: "Sequence, expression, and anti-GCRV function of the ferritin from the grass carp, Ctenopharyngodon idellus"
The findings of your research is summarised as:
"The results of the present study provide sequence, epigenetic modification and expression, and anti- GCRV functional information of Ciferritin. These provide a basis for achieving resistance to GCRV in grass carp breeding."
Please find below my comments / suggestions:
1. In your manuscript you stated: "the full-length 88 cDNA and promoter sequences of Ciferritin were identified". Do you claim this is done for the first time? Please clarify.
2. In the Line number 287, the paragraph starting with: "Taken together, ..." can be better presented under the section: Conclusion.
3. In Line number 9, the segment: " played in grass carp " need to be re-written.
4. The term "CiFerritin" is mostly typed with small "f". please keep consistency. I wonder why this term is not located in the manuscript title?
5. The last sentence in the Abstract needs correction: "which provides a basis for achieving resistance to GCRV " (conside using "provide" instead of "provides". The subject of the sentence is "plural").
6. Sentence in Lines 83-85 "immune molecule and clarify its anti-GCRV function is of significance" needs to be re-written.
7. The quality of Figures need to be improved. It seems that some Figures (e.g. Figure 3 and Figure 4) need Copyright permission.
8. The fluorescence microscopy study details are not sufficiently given. I suggest you put them under a separate Section.
Author Response
- In your manuscript you stated: "the full-length cDNA and promoter sequences of Ciferritin were identified". Do you claim this is done for the first time? Please clarify.
Answer: For both the Ciferritin middle-subunit cDNA and the promoter sequences, it is the first time to obtain these sequences in grass carp, and we clarified it in Line 205. 2. In the Line number 287, the paragraph starting with: "Taken together, ..." can be better presented under the section: Conclusion.
Answer: We changed the “Taken together” to “In conclusion”, please see Line 289.
- In Line number 9, the segment: " played in grass carp " need to be re-written.
Answer: we have re-written the segment, please see Line 9-10.
- The term "CiFerritin" is mostly typed with small "f". please keep consistency. I wonder why this term is not located in the manuscript title?
Answer: Thanks for your valuable comments, we have checked the whole manuscript and changed the "CiFerritin" to "Ciferritin". The "Ciferritin" is the abbreviation form of "ferritin from Ctenopharyngodon idellus", and in the title, we used the full name of "ferritin from Ctenopharyngodon idellus" to make the reader easy to understand, thus we only used the "Ciferritin" in the text.
- The last sentence in the Abstract needs correction: "which provides a basis for achieving resistance to GCRV " (consider using "provide" instead of "provides". The subject of the sentence is "plural").
Answer: we have changed “provides” to “provide”, please see Line 30.
- Sentence in Lines 83-85 "immune molecule and clarify its anti-GCRV function is of significance" needs to be re-written.
Answer: we have re-written the sentence, please see Line 85-86.
- The quality of Figures need to be improved. It seems that some Figures (e.g. Figure 3 and Figure 4) need Copyright permission.
Answer: Thanks for the suggestions. We will improve the quality of figures as the requirement of the press. Figure 3 is the tertiary structure of ferritin proteins we predicted by the Simple Modular Architecture Research Tool, and Figure 4 is a Phylogenetic tree we constructed by MEGA 7.0 software. There is no requirement for Copyright permission.
- The fluorescence microscopy study details are not sufficiently given. I suggest you put them under a separate Section.
Answer: We have added a separate section of the fluorescence microscopy study as the reviewer’s suggestions, please see Line 426-432.